

# High prevalence of *Lynx rufus* gammaherpesvirus 1 in wild Vermont bobcats

Dagan A. Loisel[1], Ryan M. Troyer[2] and Sue VandeWoude[3]

[1] Department of Biology, St. Michael's College, Colchester, VT, USA
[2] Department of Microbiology & Immunology, University of Western Ontario, London, ON, Canada
[3] Department of Microbiology, Immunology, and Pathology, Colorado State University, Fort Collins, CO, USA

## ABSTRACT

Gammaherpesviruses (GHVs) are host specific DNA viruses that infect a large range of mammalian species. These viruses preferentially target host lymphocyte cell populations and infection may lead to morbidity or mortality in immunocompromised, co-infected, or non-adapted hosts. In this study, we tested for the presence of *Lynx rufus* gammaherpesvirus 1 (LruGHV1) in a northeastern United States population of wild bobcats (*L. rufus*). We estimated prevalence of infection and viral load in infected individuals using quantitative real-time PCR analysis of spleen DNA from 64 Vermont bobcats. We observed an overall prevalence of 64% using this methodology. Bobcat age was significantly positively associated with GHV infection status, and we noted a trend for higher viral loads in young animals, but prevalence and viral load were similar in male and female bobcats. A single LruGHV1 variant was identified from the sequencing of the viral *glycoprotein B* gene of Vermont bobcats. This gene sequence was 100% similar to that reported in Florida bobcats and slightly variant from other isolates identified in the Western USA. Our work suggests broad geographic distribution and high prevalence of LruGHV1 in bobcat populations across the United States with infection attributes that suggest horizontal transmission of the agent. Geographic differences in viral genotype may reflect historical migration and expansion events among bobcat populations.

# INTRODUCTION

Herpesviruses are large, enveloped, double-stranded DNA viruses capable of both lytic (i.e., productive) and latent infection of their hosts (*Speck & Ganem, 2010*). This ability to express true latency results in lifelong infection punctuated by episodes of reactivation and productive infection in response to environmental stimuli. Herpesviruses are classified into three subfamilies—alpha, beta, and gamma—based on differences in genomic sequence, tissue tropism, host range, initial form of infection, and other biological properties. Gammaherpesviruses (GHVs) are highly host specific and typically cause

Corresponding author
Dagan A. Loisel, dloisel@smcvt.edu

persistent, inapparent, latent infections in lymphocytes (*Speck & Ganem, 2010*). GHV-associated disease is primarily observed in immunocompromised hosts as a consequence of latent infection, and to a lesser extent, reactivation of lytic infection (*Ackermann, 2006*; *O'Toole & Li, 2014*). In humans, latent infection with GHVs, such as Epstein–Barr virus (EBV) and Karposi sarcoma-associated herpesvirus, may lead to a variety of lymphoproliferative disorders (LPDs) in individuals with impaired immunity due to HIV infection and AIDS, therapeutic immunosuppression, or congenital immunodeficiency (*Cesarman, 2011*, *2014*).

In animals, GHV infection is generally asymptomatic when found in its natural host, but can lead to pathogenic LPDs in the case of host immune dysfunction, immunodeficiency, and/or coinfection with another virus (*Ackermann, 2006*; *Kaye et al., 2016*). In domestic cats, the probability of GHV detection and associated viral load was higher in cats infected with the pathogenic lentivirus, feline immunodeficiency virus, or in cats that were presented for other illnesses (*Beatty et al., 2014*; *Ertl et al., 2015*; *Tateno et al., 2017*). In addition, GHV transmission to a heterologous or non-adapted host may result in disease. For example, cattle exhibit the fatal lymphoproliferative disease, malignant catarrhal fever, as a consequence of infection with wildebeest or sheep GHVs that cause inapparent infection in their natural hosts (*Russell, Stewart & Haig, 2009*).

Recently, novel GHVs were discovered circulating in North American populations of two wild felids, bobcats (*Lynx rufus*) and pumas (*Puma concolor*) (*Lozano et al., 2015*; *Troyer et al., 2014*). Two strains, *Lynx rufus* gammaherpesvirus 1 (LruGHV1) and LruGHV2, were identified in wild bobcats, and interestingly, the common bobcat strain (LruGHV1) was also detected in wild pumas, in addition to a strain that was puma-specific (PcoGHV1) (*Troyer et al., 2014*). Furthermore, five polymorphic sites in the LruGHV1 *glycoprotein B* (*gB*) gene were observed in comparisons of sequences from Florida and California bobcats, suggesting population-specific differences in viral evolution (*Troyer et al., 2014*). In the same study, a third felid GHV, FcaGHV1, was identified in North American domestic cats (*Felis catus*) but not wild felids (*Troyer et al., 2014*). Subsequent studies documented FcaGHV1 DNA in cats sampled in Australia, Europe, Asia, and the US, suggesting a worldwide distribution of this potentially pathogenic virus (*Beatty et al., 2014*; *Ertl et al., 2015*; *McLuckie et al., 2016*; *Tateno et al., 2017*). Interestingly, the vast majority of domestic cats infected with FcaGHV1 have been male (*Beatty et al., 2014*; *Ertl et al., 2015*; *McLuckie et al., 2016*; *Tateno et al., 2017*). Serologic analysis determined that approximately half of domestic cats that harbored FcaGHV1 specific antibodies did not have detectable DNA in circulation, suggesting that prevalence estimates based on the detection of FcaGHV1 DNA may be understating the extent of infection (*Stutzman-Rodriguez et al., 2016*). A causal role for FcaGHV1 in the development of lymphomas or other diseases in domestic cats has been hypothesized, but not yet demonstrated.

In this study, we investigated the presence and prevalence of GHV in a wild population of bobcats in the northeast United States. We used quantitative real-time PCR (qPCR) to detect GHV DNA in spleen samples from Vermont bobcats in order to estimate prevalence of GHV infection in the population and viral load in infected individuals. We then
sequenced a segment of the GHV *gB* gene in infected bobcats to determine GHV strain identity and to characterize potential population-specific genetic variants. Finally, we tested for an association between the bobcat demographic traits, age and sex, and viral infection status to better understand the transmission and ecology of the virus.

## MATERIALS AND METHODS

### Sample collection

The protocol to collect animal samples was approved prior to collection by the Saint Michael's College Institutional Animal Care and Use Committee under protocol #022-2016. Spleen samples were obtained from bobcat carcasses collected by the State of Vermont Fish and Wildlife Department during the 2015 (1/29/15–2/12/16) and 2016 (3/12/16–2/5/17) seasons. Carcasses showing severe autolysis were not sampled. Immediately after collection, spleen samples were assigned unique alphanumeric identifiers, transferred to cryogenic storage tubes containing RNA*later* stabilization solution (Ambion by Life Technologies, Carlsbad, CA, USA), and stored at −20 °C temporarily, and at −80 °C permanently in the Biology Department at Saint Michael's College. Spleen tissue was chosen for study in order to maximize the probability of detecting GHV based on previous studies documenting the highest prevalence in spleen compared to other tissues and blood (*Beatty et al., 2014*; *Lozano et al., 2015*) and the observation that GHVs are more readily detected during latency in organs, such as the spleen, containing a significant B lymphocyte cell population (*Coleman, Nealy & Tibbetts, 2010*).

Information about bobcat age, sex, and approximate location of collection (i.e., city or town) was provided by the Vermont Fish and Wildlife Department. Age of individual bobcats was estimated from a species-specific standardized aging model based on the number of cementum annuli observed in Giemsa-stained histological sections of a lower canine tooth, as performed by Matson's Laboratory (Manhattan, MT, USA) using methodology described previously (*Boertje, Ellis & Kellie, 2015*). Age categories were defined as follows: young bobcats correspond to age estimates up to two years of age and adult bobcats correspond to age estimates greater than or equal to two years of age. Spleen samples from the 2015 and 2016 seasons that had associated age, sex, and collection location data were included in this study. Of the 64 samples meeting that criteria, 48.4% were derived from bobcats harvested by licensed trappers, 37.5% were derived from bobcats harvested by licensed hunters, 12.5% were collected from dead bobcats after motor vehicle collisions, and 1.6% had unknown provenance, according to the Vermont Fish and Wildlife Department.

### DNA extraction and quantification

Total DNA was extracted from spleen tissue samples (~400 mg each) using the manufacturer's recommended protocol for the Gentra Puregene Tissue Kit (Qiagen, Inc., Valencia, CA, USA). After extraction, the concentration (ng/µl) and purity (260/280 ratio) of DNA in each sample was determined using a NanoDrop 2000 spectrophotometer (Thermo Fisher Scientific, Wilmington, DE, USA). DNA samples were diluted to 100 ng/µl working stocks for subsequent genetic analyses.

## Quantitative real-time PCR

The *gB* gene of LruGHV1 was targeted for amplification using primers and 5′ 6-carboxyfluorescein (FAM) labeled and 3′ 6-carboxytetramethylrhodamine (TAMRA) labeled probe sets as described previously (*Troyer et al., 2014*). Reactions were run using iTaq universal probes Supermix (Bio-Rad, Hercules, CA, USA), 400 nM primers, 200 nM probe, and 500 ng of template spleen DNA in a total volume of 25 μl. Cycling conditions consisted of an initial 95 °C step for 3 min, followed by 45 cycles of 95 °C for 5 s and 62 °C for 30 s. Reactions were run in a 96-well format on a CFX Connect real-time system (Bio-Rad, Hercules, CA, USA). LruGHV1 plasmid standards for quantitation were prepared by cloning the portion of the *gB* gene amplified by the degenerate pan-GHV polymerase chain reaction (PCR) into pCR4-TOPO using a TOPO TA cloning kit (*Troyer et al., 2014*). Dilutions of plasmids ranging from $10^7$ to $10^2$ copies per reaction mixture were prepared in a background of salmon sperm DNA equivalent to 250 ng per reaction mixture, and these standards were run in duplicate in all qPCR runs. Amplification efficiency ranged between 95% and 103.6% for all qPCR runs. Spleen DNA samples were tested in duplicate, and an individual sample was considered positive only if both replicates were positive with greater than three copies per reaction. For quantitation of the number of copies per million cells, the number of cell equivalents for each DNA sample was determined as described previously (*Terwee et al., 2008*; *Troyer et al., 2014*).

## PCR amplification and sequencing of *gB* gene

A nested PCR amplification of the gammaherpesvirus *gB* gene was performed using the iProof High Fidelity DNA polymerase system (Bio-Rad, Hercules, CA, USA) on a T100 Thermal cycler (Bio-Rad, Hercules, CA, USA). In the first round, the PGHV-F5 forward (GGGGATGTGATTTCGGTGAC) and PGHV-R5 reverse (TCGACCACCTCAAAGTCAATG) primers were used to amplify a 368 bp region of the *gB* gene from spleen DNA samples. Each reaction contained 100 ng of spleen DNA in a 50 μl reaction of 1× iProof HF Buffer, 200 μM dNTP mix, 0.5 μm PGHV-F5 forward primer, 0.5 μm PGHV-R5 reverse primer, 1.0 mM MgCl$_2$, 0.5 units of iProof DNA polymerase, and DNAse/RNAse-free water. In the second round, the PGHV-F6 forward (GCATGAGAGTTCCAGGTCCA) and PGHV-R6 reverse (TGATGAAGGTGTTGAGAGTTGAAA) primers were used to amplify a 258 bp region of the PCR product from round one. Each reaction contained 2 μl of round one PCR product in a 50 μl reaction of 1× iProof HF Buffer, 200 μM dNTP mix, 0.5 μm PGHV-F6 forward primer, 0.5 μm PGHV-R6 reverse primer, 1.0 mM MgCl$_2$, 0.5 units of iProof DNA polymerase, and DNAse/RNAse-free water. The cycling conditions for both rounds were as follows: 98 °C for 2 min, followed by 35 cycles of 98 °C for 30 s, 68 °C for 30 s, and 72 °C for 30 s, with a final extension step of 72 °C for 5 min. PCR product (4 μl) from round two was run on a 2% TAE agarose gel stained with Gel Red DNA dye (Biotium, Inc., Fremont, CA, USA) and visualized under ultraviolet light.

For PCR products showing a single, clear band of the expected size after the two rounds of amplification, a single-step enzymatic cleanup was used to eliminate unincorporated primers and dNTPs prior to sequencing (ExoSAP-IT for PCR

Product Cleanup; Affymetrix, Santa Clara, CA, USA). Samples were shipped to Macrogen Corporation (Rockville, MD, USA) for single primer extension, dye-terminator, capillary sequencing on an ABI 3730XL DNA Analyzer (Applied Biosystems, Foster City, CA, USA). Each sample was sequenced in both directions and sequencing results were aligned using MEGA7 version 7.0.14 (*Kumar, Stecher & Tamura, 2016*). Consensus *gB* sequences for each Vermont bobcat were compared to *gB* sequences obtained previously from Florida, Colorado, and California bobcats (*Troyer et al., 2014*).

## Data and statistical analysis

Overall, sex-specific, and age-specific GHV prevalence estimates were calculated by dividing the total number of bobcats with spleen samples containing GHV DNA, as assessed by the *gB* qPCR assay, by the total number of bobcats tested. The 90% confidence intervals (CI) for these prevalence estimates were calculated using JMP 10.0.0. The statistical significance of sex differences in GHV prevalence was assessed using counts of positive and negative animals of each sex in $2 \times 2$ contingency tables assessed by a Pearson Chi-square test, as implemented in JMP 10.0.0. Statistical significance of differences in GHV prevalence between age categories and between seasons of collection were assessed using the Pearson Chi-square test in JMP 10.0.0. Differences in the median viral load between the sexes and age groups were assessed with the non-parametric Wilcoxon rank sum test in JMP 10.0.0.

## RESULTS

Spleen DNA samples from 64 Vermont bobcats were tested for the presence of GHV DNA using a qPCR assay targeting the viral *gB* gene. Overall, 64.1% (90% CI [53.8–73.2]) of the 64 bobcats were positive for GHV via PCR, and infected animals were identified throughout the state of Vermont (Fig. 1). After stratifying by sex, males showed a 57.1% (90% CI [41.8–71.2]) prevalence while 69.4% of females were PCR positive (90% CI [55.8–80.3]) (Table 1). This difference was not statistically significant ($\chi2 = 1.035$, d$f = 1$, $P = 0.31$). GHV detection was nearly two-fold higher in 24 adult bobcats (91.7%, CI [77.7–97.2]) compared to the 39 young bobcats (46.2%, CI [33.7–59.1]) (Table 1), and this difference in prevalence between age categories was statistically significant ($\chi2 = 13.277$, d$f = 1$, $P = 0.00027$). GHV prevalence in spleen samples collected in the 2015 season was similar to that observed in samples collected in the 2016 season ($\chi2 = 0.405$, d$f = 1$, $P = 0.52$) (Table 1).

In the 41 GHV-positive samples, the median viral load, calculated as the number of GHV DNA genomes per million host cells, was 1,590 copies/$10^6$, with a range of 52–27,931 copies/$10^6$. Male bobcats ($n = 16$) displayed a higher median viral load (2,296 copies/$10^6$) and a narrower range (293–21,081 copies/$10^6$) compared to females ($n = 25$; median: 1,329; range: 52–27,931 copies/$10^6$) (Fig. 2), but the difference between males and females was not statistically significant (Wilcoxon rank sum test, $Z = 1.35$, $P = 0.18$). Median GHV viral load was nearly two-fold greater in young bobcats ($n = 18$; 2,866 copies/$10^6$) compared to adult bobcats ($n = 22$; 1,499 copies/$10^6$) (Fig. 2). This difference was not significant at the $P = 0.05$ level, but demonstrated an interesting trend towards higher loads in younger animals (Wilcoxon rank sum test, $Z = 1.67$, $P = 0.095$).

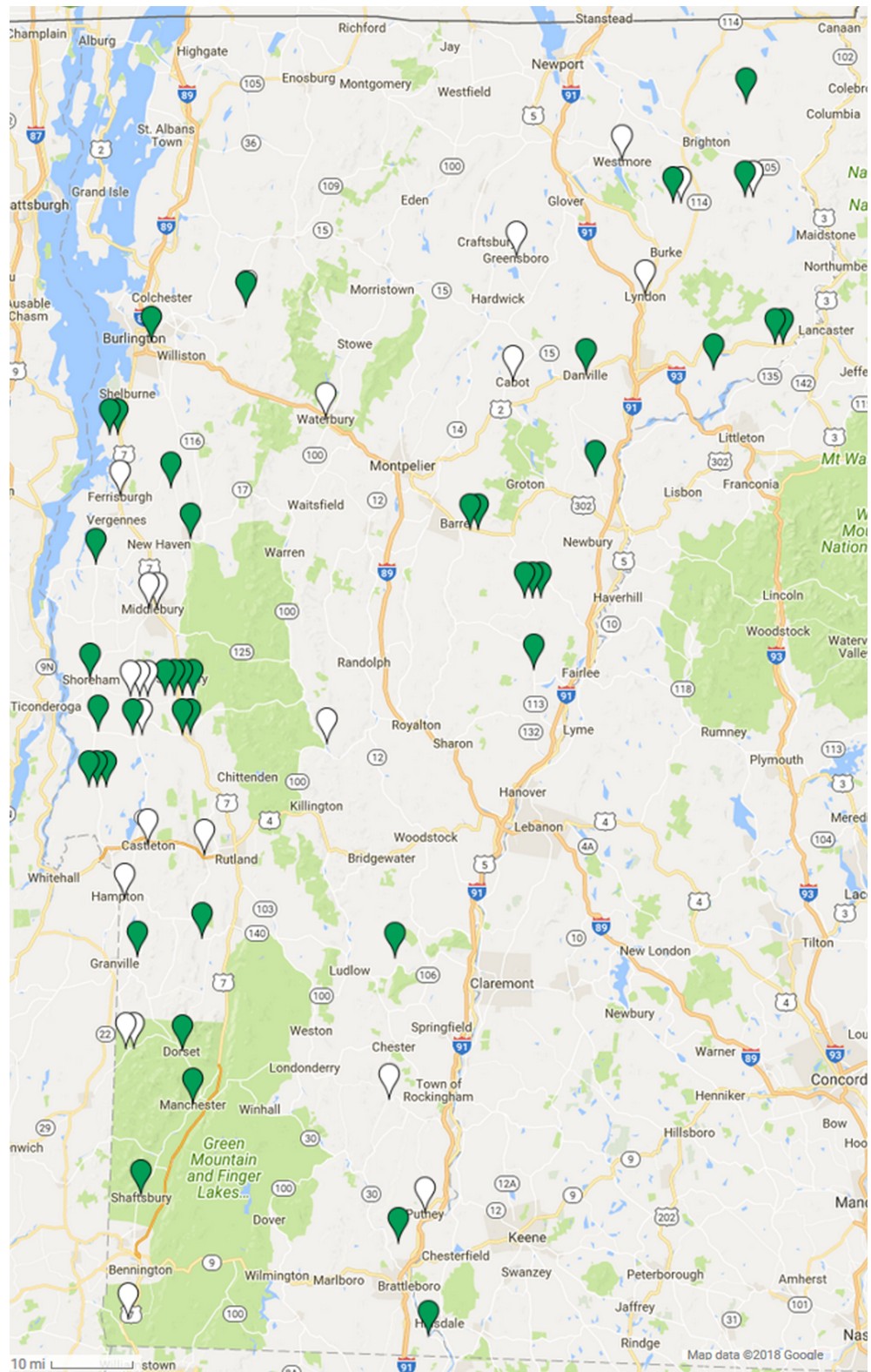

**Figure 1 Geographic distribution of gammaherpesvirus prevalence in Vermont bobcats.** LruGHV1 qPCR-positive bobcats are indicated by green pins and qPCR-negative bobcats are indicated by white pins. Map data © 2018 Google.           

**Table 1 Sample numbers and prevalence estimates from gammaherpesvirus qPCR analysis of DNA extracted from Vermont bobcat spleens.**

|  | Sample | Number of samples | | | Prevalence % | 90% CI | P value |
|---|---|---|---|---|---|---|---|
|  |  | Total | GHV negative | GHV positive |  |  |  |
| Total | All individuals | 64 | 23 | 41 | 64.06 | 53.81–73.17 |  |
| Sex | Male | 28 | 12 | 16 | 57.14 | 41.81–71.22 | 0.31 |
|  | Female | 36 | 11 | 25 | 69.44 | 55.83–80.34 |  |
| Age | Adult (≥2 years) | 24 | 2 | 22 | 91.67 | 77.69–97.20 | 0.00027 |
|  | Young (<2 years) | 39 | 21 | 18 | 46.15 | 33.70–59.10 |  |
| Season | 2015 | 30 | 12 | 18 | 60.00 | 45.01–73.29 | 0.52 |
|  | 2016 | 34 | 23 | 11 | 67.65 | 53.58–79.11 |  |

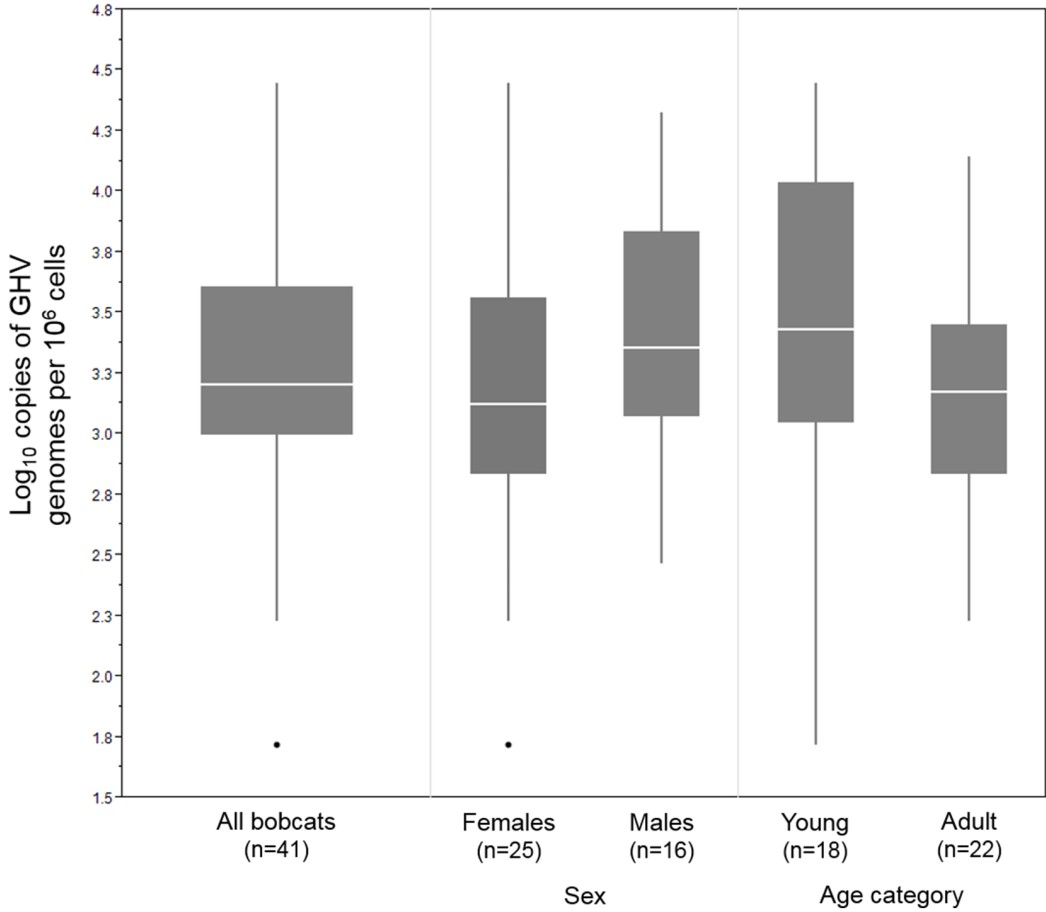

**Figure 2 DNA viral load of 41 LruGHV1-positive Vermont bobcats stratified by sex and age.** Age was not available for one GHV-positive female bobcat. Boxes show median values and 25th and 75th percentiles, while whiskers extend an additional 1.5× of interquartile range. Outliers are shown as individual points.

```
Amino acid position                          40                      50                      60
Amino acid sequence      M  R  V  P  G  P  D  N  I  C  Y  S  R  P  I  V  T  F  R  F  K  N  G  T  D  I  F  T  G
Nucleotide sequence     ATG AGA GTT CCA GGT CCA GAC AAC ATA TGC TAC TCC AGA CCC ATC GTC ACC TTC AGG TTT AAA AAC GGG ACT GAC ATC TTC ACT GGA
CA/CO isolate (x208/x262) ... ... ... ... ... ... ... ... ... ... ... ... ... ... ... ... ... ... ... ... ... ... ... ... ... ... ... ... ...
CA isolate (x159)       ... ... ... ... ... ... ... ... ... ... ... ... ... ... ... ... ... ... ... ... ... ... ... ... ... ... ... ... ...
CA isolate (x148)       ... ... ... ... ... ... ... ... ... ... ... ... ... ... ... ... ... ... ... ... ... ... ... ... ... ... ... ... ...
CA/CO isolate (x209/x356) ... ... ... ... ... ... ... ... ... ... ... ... ... ... ... ... ... ... ... ... ... ... ... ... ... ... ... ... ...
FL isolate (x1212)      ... ... ... ... ... ... ... ... ... ... ... ... ... ... ... ... ... ... ... ... ... ... ... ... ... ... ... ... ...
VT isolate (SP043)      ... ... ... ... ... ... ... ... ... ... ... ... ... ... ... ... ... ... ... ... ... ... ... ... ... ... ... ... ...

Amino acid position                          70                      80                      90
Amino acid sequence      Q  L  G  P  R  N  E  I  L  L  S  T  N  L  V  E  T  C  R  D  S  A  V  H  Y  F  Q  S  G
Nucleotide sequence     CAA TTG GGT CCT AGA AAT GAA ATT CTA TTA TCA ACA AAC TTA GTG GAG ACT TGT AGA GAC TCT GCT GTT CAC TAC TTT CAG TCA GGG
CA/CO isolate (x208/x262) ... ... ... ... ... ... ... ... ... ... ... ... ... ... ... ... ... ... ... ... ... ... ... ... ... ... ... ... ...
CA isolate (x159)        T. ... ... ... ... ... ... ... ... ... ... ... ... ... ... ... ... ... ... ... ... ... ... ... ... ... ... ... ...
CA isolate (x148)       ... ... ... ... ... ... ... ... ... ... ... ... ... ... ... ... ... ... ... ... ... ... ... ... ... ... ..C ...
CA/CO isolate (x209/x356) ... ... ... ... ... ... ... ... ... ... ... ... ... ... ... ... ... ... ... ... ... ... ... ... ... ... ... ... ...
FL isolate (x1212)      ... ... ... ... ... ... ... ... ... ... ... ... ... ... ... ... ... ... ... ... ... ... ... ... ... ... ... ... ...
VT isolate (SP043)      ... ... ... ... ... ... ... ... ... ... ... ... ... ... ... ... ... ... ... ... ... ... ... ... ... ... ... ... ...
                         [*]

Amino acid position                         100                     110                     120
Amino acid sequence      H  Q  M  H  K  Y  V  N  Y  Q  H  K  S  T  I  D  I  Q  N  F  S  T  L  N  T  F  I
Nucleotide sequence     CAT CAA ATG CAT AAG TAT GTA AAC TAT CAA CAT AAA AGC ACA ATA GAT ATT CAG AAT TTT TCA ACT CTC AAC ACC TTC ATC
CA/CO isolate (x208/x262) ... ... ... ... ... ... ... ... ... ... ... ... ... ... ... ... ... ... ... ... ... ... ... ... ... ... ...
CA isolate (x159)       ... ... ... ... ... ... ... ... ... ... ... ... ... ... ... ... ... ... ... ... ... ... ... ... ... ... ...
CA isolate (x148)       ... ... ... ... ... ... ... ... ... ... ... ... ... ... ... ... ... ... ... ... ... ... ... ... ... ... ...
CA/CO isolate (x209/x356) ... ... ... ... ... ... ... ... ... ... ... ... ... ... ... ... .G. ... ... ... ... ... ... ... ... ... ...
FL isolate (x1212)      ... ... ... ... ... ... ... ... ... ... ... ... ... ... ... ... .C. ... ... ... ... ... ... ... ... ... ...
VT isolate (SP043)      ... ... ... ... ... ... ... ... ... ... ... ... ... ... ... ... .C. ... ... ... ... ... ... ... ... ... ...
                                                                            [M/T]
```

**Figure 3 Alignment of partial LruGHV1 *glycoprotein B* (*gB*) sequences from four US bobcat populations.** For each isolate, collection location is indicated by two letter state code and animal origin is listed in parentheses. Amino acid-changing substitutions are highlighted in gray with the resultant amino acid in brackets below the codon. The asterisk symbol (*) indicates a stop codon. Amino acid position was based on published LruGHV1 *gB* sequence from a CA bobcat (x208; GenBank accession number KF840716). Isolate sequences are available in GenBank as follows: x148, MH463439; x159, MH463440; x209, MH463441; SP043, MH463442.     

To identify the GHV strain detected in Vermont bobcats, a 258 bp region of the GHV *gB* gene open reading frame was sequenced in five GHV-positive individuals. The *gB* sequence obtained was identical in all five Vermont bobcats and was 99.6% similar (257/258 bases) at the nucleotide level to the published LruGHV1 sequence (KF840716) from a California bobcat, with the single nucleotide difference corresponding to an I > C amino acid substitution at codon 108 (Fig. 3). The *gB* sequence from the Vermont bobcats was an exact match to that described previously in Florida bobcats but differed from CA and CO isolates by one or two nucleotides (Fig. 3) (*Troyer et al., 2014*).

## DISCUSSION

In this study, we detected the presence of gammaherpesvirus DNA in 64% of the Vermont bobcat spleen DNA samples studied using a sensitive qPCR assay. Sex was not a significant predictor of GHV infection status in our study nor in a previous study of bobcats (*Troyer et al., 2014*), which is consistent with recent findings that sex of wild and domestic cats was inconsistently predictive of exposure to directly transmitted pathogens but not vector or environmental pathogens (*Carver et al., 2016*). This finding is in stark contrast to findings in domestic cats, which with the exception of cats in Singapore, have shown a significantly higher prevalence in male versus female cats (*Beatty et al., 2014*; *Ertl et al., 2015*; *McLuckie et al., 2016*).

A significant positive association between age and GHV prevalence was observed in Vermont bobcats, which is consistent with previous observations in bobcat populations in California, Colorado, and Florida (*Troyer et al., 2014*). In addition to GHV, this pattern of higher rates of infection in older animals has been observed for directly transmitted

(e.g., feline immunodeficiency virus), vector-borne (e.g., *Bartonell*a sp.), and environmentally transmitted (e.g., *Toxoplasma gondii*) pathogens in wild felids (*Bevins et al., 2012*; *Carver et al., 2016*). This finding may reflect the cumulative number of potential exposure events experienced by older bobcats during their lifetimes, and is highly suggestive that LruGHV1 is transmitted horizontally during adulthood and not vertically or during the perinatal period.

In contrast to the higher risk of infection in older bobcats, we observed a trend towards a higher viral load in young bobcats, which may reflect the timing of the initial infection event, the transition to latency by GHV, or developmental differences in immunocompetency. In human EBV infection, viral load in peripheral B cells peaks during primary infection, rapid declines after host immune response, and continues a slow decline until reaching the low levels typical of latency (*Fafi-Kremer et al., 2005*; *Hochberg et al., 2004*). A similar pattern was observed in viral load in the spleens of mice infected with murine GHV68 (*Olivadoti et al., 2007*). Assuming that younger bobcats had more recently experienced primary infection than older ones, higher viral loads would be expected in the younger versus older individuals based upon these observations in other GHV infections. Our observed trend of higher viral load in younger bobcats should be interpreted with caution as bobcats were sampled without regard to age in this study. To confirm the trend, future studies should be designed to explicitly compare viral loads in animals of known age, sex, and cause of death, so as to minimize the effect of confounding variables, such as autolysis or unbalanced sampling.

Identification of LruGHV1 in a northeastern US bobcat population adds to previous reports of PCR detection of LruGHV1 in blood of western (Colorado and California) and southeastern (Florida) bobcat populations (*Troyer et al., 2014*), and PCR detection of LruGHV1 from spleen DNA from Colorado (*Lozano et al., 2015*). We confirm a widespread viral range across the continental United States, with the highest PCR prevalence estimates being observed in eastern US populations, Vermont (64%) and Florida (76%). Prevalence based upon PCR analysis of blood was 25% in Colorado and 37% in California (*Troyer et al., 2014*). In contrast, PCR analysis of Colorado bobcat spleen DNA (*Lozano et al., 2015*) produced a prevalence estimate (62%) more similar to that detected in Vermont, indicating that analysis of spleen DNA may provide a more sensitive substrate for PCR detection, an observation supported by research showing that murine GHV68 latency is preferentially maintained in splenic B cells (*Flano et al., 2002*). The prevalence estimate in this study may underestimate actual levels of infection in Vermont bobcats. First, it is feasible that serologic evaluation of LruGHV1 would document an even greater prevalence of this infection in wild populations, given that FcaGHV1 PCR analysis of domestic cats only detected viral genomes in half of the cats that had seroreactivity against FcaGHV1 antigens (*Stutzman-Rodriguez et al., 2016*). Second, although severely autolyzed carcasses were not included in this study, some DNA degradation may have occurred due to autolysis prior to sample collection and storage at −80 °C resulting in a lower probability of qPCR detection of LruGHV1.

The partial *gB* gene sequence obtained from Vermont bobcats was identical to that observed from Florida bobcats and closely related to western isolates that have been

examined. This preliminary data shows more similarity among LruGHV1 isolates across eastern populations, and suggests that this pathogen may be useful to resolve geographic movements and population refugia, similar to the use of feline immunodeficiency virus sequence as a marker for these variables (*Antunes et al., 2008*; *Fountain-Jones et al., 2017*; *Lee et al., 2012*). However, this preliminary interpretation is based solely on patterns observed in a 258 bp region of the LruGHV1 *gB* gene and needs to be confirmed by sequencing more of the GHV genome in a larger number of bobcats from the different US populations.

The current GHV variant distribution reflects the expansion of bobcat populations from separate Pleistocene refugia in the southeast and northwest regions of the United States. Eastern and western US bobcat populations show strong genetic differentiation in mitochondrial and microsatellite markers which is thought to be the result of expansion from these refugia (*Croteau et al., 2012*; *Reding et al., 2012*). Thus, the GHV isolates infecting current bobcat populations may be the result of adaptation to the unique genetic structure of the eastern and western populations, or may reflect the historical migration and range expansion events. Since GHV is apparently present in a much wider range of individuals than feline immunodeficiency virus (which has only been reported in Florida and California bobcats), it may provide an excellent marker to further explore the relationship of populations across bobcat natural ranges.

## CONCLUSIONS

We document for the first time the presence of gammaherpesvirus in a population of wild felids living in the northeast United States. The high prevalence of the LruGHV1 strain in Vermont bobcats, combined with previous reports of the strain in western and southeast populations, suggests that the virus has a broad geographic distribution across the continental US. Our results show that older bobcats are more likely to be infected, but may potentially harbor lower viral loads, than young bobcats. We confirm that viral infection status is not associated with sex of bobcats, in contrast to the male-biased pattern of infection observed in domestic cats. These results suggest that horizontal transmission during adulthood is the most likely mechanism of gammaherpesvirus spread in wild bobcats. The sequence differences observed in the GHV *gB* gene in different bobcat populations may reflect historical migration and range expansion events, but this hypothesis should be tested by analyzing more of the GHV genome in a larger number of infected individuals.

## ACKNOWLEDGEMENTS

We thank the Vermont Department of Fish and Wildlife, Vermont hunters and trappers, and the Saint Michael's College Biology Department, Office of the Dean of the College, and Office of the Vice President of Academic Affairs for support of the project. We also thank the Saint Michael's College fall 2017 BI-365 class for helpful discussions. The contents of this publication are solely the responsibility of the authors and do not necessarily represent the official views of the NIGMS or NIH.

### Funding

This work was supported by a Research Opportunity Award to Dagan A. Loisel connected to a grant from the NSF-NIH Ecology of Infectious Disease program (NSF EID 1413925) to Sue VandeWoude. It was also supported by a Morris Animal Foundation (D14FE-301) grant to Ryan M. Troyer. Finally, research reported in this publication was supported by a grant to Dagan A. Loisel from an Institutional Development Award (IDeA) from the National Institute of General Medical Sciences of the National Institutes of Health under grant number P20GM103449. The funders had no role in study design, data collection and analysis, decision to publish, or preparation of the manuscript.

### Grant Disclosures

The following grant information was disclosed by the authors:
Research Opportunity Award.
NSF-NIH Ecology of Infectious Disease program: NSF EID 1413925.
Morris Animal Foundation: D14FE-301.
Institutional Development Award (IDeA).
National Institute of General Medical Sciences of the National Institutes of Health: P20GM103449.

### Competing Interests

The authors declare that they have no competing interests.

### Author Contributions

- Dagan A. Loisel conceived and designed the experiments, performed the experiments, analyzed the data, contributed reagents/materials/analysis tools, prepared figures and/or tables, authored or reviewed drafts of the paper, approved the final draft.
- Ryan M. Troyer conceived and designed the experiments, contributed reagents/materials/analysis tools, authored or reviewed drafts of the paper, approved the final draft.
- Sue VandeWoude conceived and designed the experiments, contributed reagents/materials/analysis tools, authored or reviewed drafts of the paper, approved the final draft.

### Animal Ethics

The following information was supplied relating to ethical approvals (i.e., approving body and any reference numbers):

The protocol to collect animal samples was approved prior to collection by the Saint Michael's College Institutional Animal Care and Use Committee under protocol #022-2016.

### DNA Deposition

The following information was supplied regarding the deposition of DNA sequences:

NCBI accession numbers: x148, MH463439; x159, MH463440; x209, MH463441; SP043, MH463442. Sequence data can also be found in the Supplemental Information.

## Data Availability

FASTA sequences of partial LruGHV1 glycoprotein B sequences from US bobcat populations are available in the Supplemental File.

## Supplemental Information

Supplemental information for this article can be found online at http://dx.doi.org/10.7717/peerj.4982#supplemental-information.

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
