# Peer review of "High prevalence of Lynx rufus gammaherpesvirus 1 in wild Vermont bobcats"

_PeerJ, doi:10.7717/peerj.4982_

## Round 0.1 · original submission · Minor Revisions

Please consider the corrections proposed by the referees to make the study design more clear, specially those regarding sampling and other aspects of the methods.

·

Basic reporting

The article is well written, and well referenced.

Experimental design

The experimental design is straightforward and the cases sufficient for significance. For both wildlife health, and for a relatively new and unknown virus, I laud the authors for the large study group required for the importance of the results.
A few specific questions that relate to interpretation of the results:
1. The authors should address the choice of spleen. If it is serving as an ersatz for blood in this case then that needs to be justified (what is the correlation in other studies?). I can think of reasons why that would not be correlative, and couldn't find any data about the case for FcGHV.
2. There is no description of how spleens were collected or stored. The inclusion criteria for these wildlife samples should be added.
3. The "trend" for higher loads in younger animals has to be assessed in the context of sample choice and autolysis.

Validity of the findings

With the caveats stated above in experimental design, the study is useful and well organized.
Presence and prevalence are solid.
The significance of the genetic variants is still unclear (to me, as a lay reader) and should could be stated in the conclusions (unless I am simply missing the boat). Do you experts believe that the number of field samples of the variants clearly establishes criteria for significant strain variation? Is the jury still out on potential for viral evolution? This is one of the most interesting questions, I think, because if "species jumping" (if that can be defined) in cats is associated with viral impact or virulence, then these studies could be predictive of important species crowding problems and conservation issues.

Additional comments

Minor comments:

54: "....may result in pathology". Pathology is the study of disease, so rather GHV may result in disease, but not in "pathology".
71-74: "Serologic analysis determined that approximately half of domestic cats
that harbored FcaGHV1 specific antibodies did not have detectable DNA in circulation,suggesting that approximately 30% of cats across the US have been exposed to FcaGHV" This is a non-sequitor unless I am missing the point?
82-84: Not sure that the statement "...this is the first attempt to document the presence of GHVs in wild or domestic cats in the Northeast United States" is necessary. The significance of the data stands on its own, and since your own previous studies made conclusions about the presence of the virus in wildlife and with widespread geographic presence...in my opinion this sentence dilutes rather than strengthens the impact of the study, which is clearly stated in the sentence before.

Reviewer 2 ·

Basic reporting

see below

Experimental design

see below

Validity of the findings

see below

Additional comments

High prevalence of Lynx rufus gammaherpesvirus 1 in wild Vermont bobcats (#23555)

The manuscript is very well written. The content is novel, clear and well executed. The manuscript will be of interest to virologists, particularly those interested in gammaherpesviruses, as well as wildlife ecologists.
I have the following specific comments:
Line 35-36: Ultimately, the frequency and physiological locations of reactivation determine the clinical manifestation of herpesvirus infection (Speck & Ganem 2010).
Please reconsider the accuracy of this statement with regard to gammaherpesviruses. Upon infection of the host, GHVs usually enter latency immediately and most GHV disease is manifest during viral latency. Similarly lines 42-43,

line 20: “Intensity…..of infection” is used to refer to virus load. This terminology may be confusing because it is not commonly used in this context, in my experience. Similarly line 51, 78
Line 89: Re bobcat carcasses, a note of explanation to the reader as to their provenance would assist the international audience. Did these animals die of natural causes? Were they hunted? trapped? This information is important to understand how representative the sampled population is of the general population.
Line 97-98. Please comment and reference the accuracy of this method of aging. A caveat to the interpretation of results, as they relate to age might be considered.
Line 156 and throughout manuscript qRT-PCR should be replaced with qPCR. The latter is accepted to refer to real-time quantitative PCR. (The addition or RT is unnecessary and can suggest a reverse transcription step, which is not the case)
Lines 237-242. This information is interesting and informative. However, the limitations associated with analysis of such a short sequence should be acknowledged.
Lines 249-252 Fascinating concept.

---

## Round 0.2 · accepted · Accept

The article describes novel and relevant information, and it will of course be of interest for those studying herpesviruses as well as for people dealing with the conservation of wild felids.

#